# Study on the Migration Behaviors of Magnesium Oxysulfate Nano-Whiskers in Polypropylene Composites with Surface Modification

**DOI:** 10.3390/ma16175899

**Published:** 2023-08-29

**Authors:** Jong Sung Won, Jeong Jin Park, Eun Hye Kang, Min Hong Jeon, Miyeon Kwon, Seung Goo Lee

**Affiliations:** 1Defense Materials & Energy Technology Center, Agency for Defense Development, Daejeon 34060, Republic of Korea; jswon@add.re.kr; 2Department of Organic Materials Engineering, Chungnam National University, Daejeon 34134, Republic of Korea; derinn@cnu.ac.kr; 3Department of Advanced Organic Materials Engineering, Chungnam National University, Daejeon 34134, Republic of Korea; ehkang1257@o.cnu.ac.kr (E.H.K.); hong831@kitech.re.kr (M.H.J.); 4Material and Component Convergence R&D Research Group, Korea Institute of Industrial Technology, Ansan 15588, Republic of Korea; mykwon@kitech.re.kr

**Keywords:** compounding, coating, surface modification, filler

## Abstract

In this study, surface modification aimed to enhance the compatibility between a hydrophilic inorganic filler and polypropylene (PP) matrix using hydrophobic treatment. Lauric acid, butyl acrylate, and maleic anhydride were employed to modify the filler surface. After treatment, inorganic filler/PP composites were produced using melt-mixing and extrusion–injection molding processes. The study focused on investigating compatibility and migration behavior between the filler and matrix. The findings indicated that hydrophobic modification, specifically with butyl acrylate and maleic anhydride, improved migration issues in nano-whisker, while maintaining favorable mechanical properties even under accelerated thermal aging. However, excessive hydrophobicity induced by superhydrophobic treatment using lauric acid led to reduced compatibility with the matrix, compromising its effectiveness. Consequently, the study revealed the potential of surface modification to enhance interfacial properties and mitigate migration concerns in PP composites for automotive applications.

## 1. Introduction

Composites are produced by blending fillers into a polymer matrix to enhance the mechanical strength and modulus of the matrix [1,2,3,4]. Among these composites, polypropylene (PP) is widely regarded as the predominant polymeric material owing to its versatile attributes, including its light weight, cost effectiveness, heat resistance, favorable mechanical properties, processability, chemical resistance, and transparency [5]. PP finds extensive application across various industries, particularly in automotive manufacturing, where it is utilized for both interior and exterior components, such as panels, bumpers, and door trims. To achieve desired performance characteristics for specific applications, PP is processed into compounds by incorporating various inorganic fillers, including talc, nano-whisker, glass fiber, and glass wool [6,7]. 

These inorganic fillers exert a substantial influence on mechanical and chemical properties, encompassing tensile strength, impact resistance, flexibility, elasticity, heat resistance, and dimensional stability, thus enhancing the mechanical strength and modulus of the matrix. The content of inorganic fillers is typically incorporated within a weight percentage range of 10% to 40%, depending on the targeted application [8,9,10,11]. Notably, nano-whisker, characterized by its elongated whisker-like acicular structure and high aspect ratio, enhances the melt flow of the polymer matrix, thereby reducing injection cycle times, increasing production efficiency, and facilitating weight reduction due to its low specific gravity. Consequently, nano-whisker has gained widespread acceptance as a preferred inorganic filler in polypropylene compounding [12]. 

However, following the extrusion and injection processes of PP compounds, a phenomenon known as blooming or whitening occurs, wherein inorganic fillers migrate to the surface [13,14,15]. This migration manifests as the appearance of a white residue. The primary cause of this phenomenon can be attributed to the reduced compatibility resulting from the disparity in surface energy between the hydrophilic nature of the inorganic fillers and the hydrophobic characteristics of the PP polymer matrix [2,16,17,18]. 

Additionally, the issue of discoloration formation due to external contamination persists in the polypropylene used for automotive interiors [19]. More recently, external contaminants such as cosmetics and sunscreen applied by consumers have been reported as the primary sources of discoloration. Automobile interior and exterior materials are prone to discoloration throughout a 12-year average lifespan due to long-term surface contamination induced by passenger interaction. These problems not only diminish the quality of the final product but also negatively impact key quality surveys, such as Initial Quality Index (IQS) and Vehicle Dependability Study (VDS). Changes in appearance lead to reduced consumer satisfaction, making them crucial concerns within the automotive industry [20,21].

Consequently, this study aims to enhance the compatibility of hydrophilic nano-whiskers by surface-modifying them to achieve hydrophobic characteristics. This modification intends to improve the interfacial adhesion between nano-whiskers and polypropylene, the polymer matrix, during compounding. By addressing the issues of migration and discoloration caused by external contaminants, the aim is to prevent the occurrence of these problems. To assess the effectiveness of nano-whisker surface modification, polypropylene specimens compounded with the surface-modified inorganic filler were prepared. The study further examined the compatibility and migration behavior of the inorganic filler with polypropylene, as well as the discoloration behavior resulting from external contamination. This assessment was carried out using thermally accelerated aging to confirm the effectiveness of surface modification on nano-whiskers.

## 2. Experimental

### 2.1. Materials

In this research, a form of nano-whisker known as MOS (magnesium oxysulfate; Ube Material Industries; with a length of 15 μm) was employed as the inorganic filler. Surface modification was performed using butyl acrylate (Aldrich, Saint Louis, MO, USA), maleic anhydride (DAEJUNG, Dasan, Korea), sodium dodecyl sulfate (Aldrich, Saint Louis, MO, USA), and lauric acid (SAMCHUN, Pyeongtaek, Republic of Korea). The composite was prepared by incorporating surface-modified nano-whiskers into polypropylene (Block Co-PP, Daejeon, Republic of Korea). Additionally, to induce discoloration on the surface of the PP composite, sunscreen (NIVEA Protect and Refresh, SPF 30, Hamburg, Germany) was applied using a white cotton cloth provided by CKSI, with a warp and weft of 15 tex. We selected a sunscreen product containing a combination of inorganic and organic blockers. 

### 2.2. Surface Modification of Nano-Whiskers

The experimental methods for surface modification were categorized into three main approaches, and the corresponding experimental conditions are presented in Table 1.

The first method involved surface modification using butyl acrylate, with the mechanism illustrated in Figure 1. Initially, a sodium dodecyl sulfate solution (SDS solution) was prepared by stirring 15 wt% sodium dodecyl sulfate in 1100 mL of deionized water at 210 °C for 60 min until complete dissolution. This SDS solution served as an emulsion stabilizer [22]. Subsequently, the SDS solution was mixed with butyl acrylate at concentrations of 10 wt%, 20 wt%, or 30 wt%, and 330 g of nano-whisker was added to the mixture. The reaction was then maintained at 85 °C for 2 h. Following the reaction, the modified nano-whiskers were thoroughly washed with deionized water to remove any suspended emulsion. Subsequently, they were dried at temperatures of 120°C or higher and subsequently pulverized. The surface-modified nano-whiskers obtained with this method were denoted as B-10, B-20, and B-30. The concentration of butyl acrylate for this condition was determined based on a literature review [23]. 

The second experimental method involved surface modification using a combination of butyl acrylate and maleic anhydride. The mechanism is illustrated in Figure 2. To initiate the process, an aqueous solution of maleic anhydride was prepared by stirring 1 wt%, 2 wt%, or 4 wt% maleic anhydride in 1375 mL of deionized water at 65 °C for 60 min. Following the same procedure as the first experimental method, the SDS solution was prepared. Subsequently, butyl acrylate was mixed in at a concentration of 20 wt%, and 330 g of nano-whisker was added and thoroughly stirred. Within 30 min, the aqueous solution of maleic anhydride was added, and the reaction was maintained at 95 °C for 4 h. The modified nano-whiskers were then thoroughly washed with deionized water to remove any suspended emulsion, dried at a temperature of 120 °C or higher, and pulverized. The surface-modified nano-whiskers obtained with this method were named BM-1, BM-2, and BM-4. For the purpose of comparison with Method 1, a concentration of 20% butyl acrylate was employed.

The third experimental method involved superhydrophobic surface modification using lauric acid and differed from the previous two methods. The mechanism is presented in Figure 3. Initially, 330 g of nano-whiskers were mixed with deionized water to form a 7 wt% slurry, which was then dispersed in an ultrasonic bath for 1 h. The slurry was subsequently stirred at room temperature for 5 h. Next, a solution of 5 wt% lauric acid in ethanol was prepared and added to the nano-whisker slurry, followed by stirring at 90 °C for 120 min. The mixture was then cooled to room temperature. After removing the fatty acids with deionized water, the product was dried at 60 °C for at least 12 h and then pulverized. The surface-modified nano-whiskers obtained with this method were named L-5. The concentration of lauric acid for this condition was determined based on a literature review [24].

### 2.3. Preparation of Polypropylene Composites

PP composites were prepared by compounding surface-modified nano-whiskers at a content of 10 wt%. Specimens with dimensions of 100 mm × 300 mm × 3 mm were extruded using a twin-screw extruder (48 mm twin screw; Changsung P&R Co., Ltd., Ansan, Republic of Korea). The extruder operated at a feed rate of 25 rpm with side feeding to maintain the acicular shape of the nano-whiskers. The extrusion speed was set at 180 rpm, with the die temperature at 220 °C and the mixing zone temperature at 240 °C.

### 2.4. External Contamination and Accelerated Aging Experiments

For each discoloration specimen measuring 50 mm × 50 mm, a uniform application of 0.5 g of sunscreen (NIVEA Protect and Refresh, SPF 30) was made on a white cotton cloth using fingers. The cloth was then placed on the surface of the specimen. To ensure Seven distribution of the load, an aluminum plate measuring 50 mm × 50 mm was placed on top of the cloth, followed by a 500 g weight. Subsequently, thermally accelerated aging was performed at a temperature of 140 °C for 24, 48, and 72 h.

The removal of residual sunscreen formed on the surface of the PP composite material due to accelerated aging and external contamination followed the same procedure. After the accelerated aging process, the 500 g weight, aluminum plate, and white cotton cloth were removed. The specimens were then allowed to cool down at room temperature for 10 min before cleaning. A mild detergent solution, specifically a 10% dilution of Soonsaem Baking Soda and Phytoncides detergent (Aekyung Industrial Co., Ltd., Seoul, Republic of Korea) in distilled water, was used for cleaning. The first cleaning step involved thoroughly soaking a white cotton cloth in the mild detergent solution, placing it on the surface of the sunscreen-applied PP composite, and holding it for 3 min. A force of approximately 10 MPa was then applied, and cleaning proceeded at a speed of 60 times per minute. The second and third washes omitted the process of holding the white cotton cloth for 3 min. Three washes were performed per specimen following the aforementioned method. 

### 2.5. Characterization of PP Composites

To observe the morphological and chemical properties, a specimen made of PP composite, incorporating surface-modified nano-whiskers in compound form, was examined using an Ultra-High-Resolution Field-Emission Scanning Electron Microscope (JSM-7610F; JEOL, Tokyo, Japan). The observation was conducted at an accelerating voltage of 15 kV, and photographs were taken at the magnification of 5000. Contact-angle measurements (KRUSS) were performed using the sessile drop method to assess the hydrophilic and hydrophobic properties of the surface-modified nano-whiskers. Pelletized powdered nano-whiskers were used to analyze the contact angle, with distilled water being used to observe hydrophilic properties and diiodomethane being used for hydrophobic properties. The hydrophilic and hydrophobic properties were characterized by measuring the θ value, which represents the angle between the solvents used.

The chemical-structure changes of the sample surface after surface modification of nano-whiskers under each condition were determined using Attenuated-Total-Reflectance Fourier-Transform Infrared Spectroscopy (ATR-FTIR; ALPHA-P; Bruker Optics Inc., Billerica, MA, USA). The measurements were performed in the range of 4000–400 cm^−1^. X-ray Photoelectron Spectroscopy (Thermo VG Scientific; K-alpha, Waltham, MA, USA) was employed to measure the chemical composition changes on the surface and inside the samples of powdered nano-whiskers-and-PP composites resulting from surface modification. The analysis area ranged from 30 to 400 μm, with measurements being taken at 5 μm intervals.

Tensile tests were conducted according to ASTM D638 to determine the mechanical properties of the PP composites prepared using different variables. The tests were performed using a universal testing machine (Shimadzu; AG-250Knx, Kyoto, Japan). Five specimens were measured for each condition, and the average value was calculated, excluding the maximum and minimum values.

## 3. Results and Discussion

### 3.1. Characterization of Inorganic Filler after Surface Modification

#### Surface Analysis

After the surface modification of nano-whisker, an inorganic filler, an SEM analysis was conducted to observe the morphological changes on the surface before and after accelerated thermal aging. Figure 4a shows SEM images of the nano-whisker surface without surface modification. In Figure 4b–h, the surfaces of nano-whiskers under seven different surface modification conditions are displayed. The SEM analysis showed that the acicular morphology of whisker was maintained under all surface modification conditions. The acicular morphology is a very important morphological feature, as it enhances the scratch resistance of automotive interior and exterior materials and allows for thinner molded parts [25,26]. 

Figure 4i shows the EDS analysis results of the surface-modified nano-whisker surface under each condition, showing the elemental composition of the surface. The specific values can be found in Appendix A. Based on the unmodified Whisker sample, L-5 had significantly reduced Mg content. This could be attributed to a mechanism in which the hydroxyl group of lauric acid binds to the Mg on the surface of nano-whisker [24]. The samples treated with butyl acrylate alone showed a decrease in the content of Mg and S with the increase in concentration. The samples treated with maleic anhydride after butyl acrylate treatment showed no difference in concentration at 1 and 2 wt%, but the samples treated with 4 wt% showed a decrease in Mg content. Mg is the main element responsible for discoloration in the manufacturing of plastic specimens that are later compounded with polypropylene. Therefore, it was found that the content of Mg decreased due to the surface modification of inorganic filler.

The goal of this research was to improve the compatibility between the hydrophobic polypropylene matrix and the hydrophilic nano-whisker by altering the surface of the filler to obtain hydrophobicity. Contact-angle measurements using the sessile drop method were conducted to evaluate the hydrophilic and hydrophobic properties of the modified filler. Figure 5a illustrates that the superhydrophobic method using lauric acid exhibited the highest contact angle with distilled water, indicating a high level of hydrophobicity. Among the methods utilizing butyl acrylate alone, the B-30 condition at the highest concentration showed the highest degree of hydrophobicity. On the other hand, surface modification with maleic anhydride did not result in significant differences, although the BM-4 sample showed slightly lower hydrophobicity. In Figure 5b, the contact angles measured using diiodomethane were generally similar for the remaining samples, except for Whisker. The observed differences can be attributed to the analytical conditions used during the contact-angle measurements. Pelletization of the powdered samples may lead to a higher absorption rate of the measuring substance, resulting in rapid absorption. In contrast, diiodomethane, being denser than distilled water, leads to smaller droplet size and improved wettability with the hydrophobized samples.

In theory, surface modification with lauric acid, butyl acrylate, and maleic anhydride can occur via physical adsorption or chemical reactions [27]. To verify the chemical composition of the surface-modified inorganic filler, a Fourier-Transform Infrared Spectroscopy (FT-IR) analysis was conducted, and the results are presented in Figure 6. Following surface modification, a distinct and broad peak with multiple bands ranging from 3600 to 3400 cm^−1^ was observed, corresponding to the stretching vibration of the -OH groups in the modified nano-whiskers. An increased absorption peak was observed, indicating the presence of additional -OH groups. Furthermore, absorption peaks at 2960, 2860, and 1380 cm^−1^, commonly observed after surface modification, can be attributed to the asymmetric, symmetric, and symmetric strain vibrations of -CH_3_, respectively. The peaks at 2910 and 2844 cm^−1^ correspond to the asymmetric and symmetric vibrations of -CH_2_. The peak at 1730 cm^−1^ signifies the stretching vibration of C=O, while the small absorption peaks at 1620 and 1470 cm^−1^ represent the asymmetric and symmetric vibration peaks of -COO-, respectively [23,24]. 

As a result, as shown in Figure 6, the changes in peaks observed in FT-IR spectra with different surface modification methods and conditions are not significant. However, when compared with the Figure 6a graph displaying the surface without modification, it is clear that the surface-modified nano-whisker contained the aforementioned chemical composition.

Figure 7 and Appendix A demonstrate the results of the XPS analysis, which quantitatively assessed the impact of surface modification on the elemental composition. The analysis revealed a significant decrease in Mg1s content after surface modification, with the largest reduction being observed with lauric acid modification, followed by butyl acrylate and maleic anhydride modification. Conversely, the content of C1s exhibited an opposite trend, increasing with the decrease in Mg1s content. The most effective surface modification condition, BM-4, demonstrated lower Mg1s content and higher C1s content. In Figure 2, it can be observed that small polymerizable monomers coexist with Mg ions and hydroxyl groups on the surface of nano-whisker. When the initiator is introduced to the micro-emulsion, these small monomers undergo copolymerization, resulting in the formation of an organic coating on the surface of nano-whisker [23]. The findings were further confirmed with an FT-IR analysis, which identified the functional groups associated with the copolymers. Additionally, the increase in the peak height of the C1s spectrum on the modified nano-whisker surface indicated the thickness of the copolymer layer. These results provide insights into the surface modification model of nano-whisker, aligning with the elemental analysis and surface characterization outcomes.

### 3.2. Characterization of PP Composites after Thermally Accelerated Aging

#### 3.2.1. Surface Analysis

To examine the morphological characteristics of the PP composite surface after thermally accelerated aging, an SEM analysis was performed, and the results are displayed in Figure 8a–e. (SEM images of other conditions are shown in Appendix A). Figure 8a,b reveal that the PP composite solely composed of 100% PP maintained a clean surface even after thermally accelerated aging. However, the PP composite containing nano-whisker without surface modification exhibited the most pronounced migration phenomenon on the surface following thermally accelerated aging. This migration can be attributed to the high-temperature migration of the inorganic filler due to decreased compatibility with the polymer matrix [28,29,30]. The SEM analysis revealed that the inorganic filler modified with butyl acrylate and maleic anhydride exhibited the least migration. To provide a more quantitative analysis, an EDS analysis was conducted.

Figure 8f shows the findings of the EDS analysis, which show changes in the elemental composition of the composite surface as the PP composites containing the inorganic filler were thermally accelerated for 0, 24, 48, and 72 h. The exact values are summarized in Appendix A. According to Figure 8f, PP composites containing nano-whisker without surface modification increased in Mg content as the thermally accelerated aging period increased due to nano-whisker migration within the composites. However, after thermally accelerated aging, excessive hydrophobicity led to migration due to poor compatibility with the polypropylene matrix. PP composites containing inorganic fillers modified with butyl acrylate and maleic anhydride exhibited minimal changes in Mg content before and after thermally accelerated aging. This suggests that the compatibility between the inorganic filler and the polypropylene matrix increased, leading to a reduction in the migration phenomenon.

An FT-IR analysis was conducted to examine the composition changes on the surface of polypropylene specimens induced by thermally accelerated aging. The results of the FT-IR analysis on polypropylene samples before and after thermally accelerated aging are presented in Figure 9. Peaks arising from the asymmetric stretching of CH_3_ and CH_2_ groups in the PP samples are identified at 2957 and 2920 cm^−1^, while symmetric and asymmetric scissoring vibrations of the methyl group are evident at 1455 and 1376 cm^−1^. Additionally, absorption peaks related to -CH_3_ rocking vibrations appear at 972 and 997 cm^−1^ [31,32,33]. When considering the peaks induced by whisker (1170 and 1730 cm^−1^), as observed in Figure 7, it becomes apparent that the overall PP-related peaks diminish while the whisker-related peaks increase following thermally accelerated aging.

In Figure 8f, nano-whiskers modified with butyl acrylate exhibited no substantial difference in elemental Mg content following thermally accelerated aging, in comparison to unmodified nano-whiskers. This outcome was similarly corroborated by the FT-IR analysis. Conversely, in the case of nano-whiskers modified with lauric acid under the L-5 condition, an initial reduction in Mg content was noted. However, after thermally accelerated aging, the excessively hydrophobic filler displayed migration due to its reduced compatibility with polypropylene. Consequently, significant alterations were observed in both EDS and FT-IR results. Consequently, the optimal conditions for modification were determined to be those involving butyl acrylate, which exhibited improved compatibility and minimized migration issues.

#### 3.2.2. Mechanical Properties

Engineering plastics often need to withstand various pressures and impacts in practical applications, making their mechanical properties highly significant. These properties can be measured using standardized test methods, with tensile testing being a common approach for evaluating plastic properties [34]. Tensile testing involves applying an external tensile load to a specimen and assessing the relationship between stress and strain until the material fails. The results are displayed in Figure 9 (the S-S curves for this tensile experiment are shown in Appendix A).

The PP composite containing L-5 exhibited exceptionally high toughness, maintaining stress even under significant deformation without breaking. These findings indicate that the composite is resistant to impacts and repetitive loading. Furthermore, the L-5 condition exhibited superior toughness even after thermally accelerated aging. This can be attributed to the superhydrophobic treatment, which resulted in lower compatibility with the matrix and a significant loss in interfacial adhesion, resulting in internal slip during the tensile test.

Furthermore, B-30 had the lowest elongation among the specimens containing nano-whisker modified with butyl acrylate, while BM-4 had the lowest elongation among the specimens containing nano-whisker modified with maleic anhydride. This demonstrates that increased treatment material concentrations in surface modification resulted in lower elongation, indicating improved interfacial adhesion between nano-whisker and the polypropylene matrix within the composite.

Overall, when comparing the tensile test results before and after thermally accelerated aging, a decrease in elongation after the aging process was observed. This can be attributed to the development of internal defects during thermally accelerated aging. Conversely, the composites containing non-surface-modified nano-whiskers exhibited minimal differences in elongation before and after thermally accelerated aging, indicating fewer internal defects.

The tensile strength represents the maximum stress experienced by the material before failure under a tensile load. The modulus of a material shows its deformation within the elastic limit in response to stress, with a higher modulus suggesting less deformation induced by the given load. In Figure 10a, the tensile strength of PP composites before thermally accelerated aging is higher for B-10, B-20, and B-30, while the modulus of elasticity is the highest for L-5. In contrast, PP composites containing the surface-modified nano-whisker exhibited higher tensile strengths than those without surface modification after thermally accelerated aging. The elastic modulus displayed a significant increase in standard deviation, indicating the inhomogeneity of the overall specimen properties. This phenomenon is likely a result of the migration of nano-whisker and its uneven distribution at high temperatures.

## 4. Conclusions

This study aimed to address the compatibility challenges arising from the hydrophobic nature of polypropylene (PP) matrix and the hydrophilic nano-whisker by modifying the surface of nano-whisker to become hydrophobic. We conducted contact-angle measurements using the sessile method to evaluate the hydrophilic and hydrophobic properties of the surface-modified nano-whisker. The superhydrophobic method utilizing lauric acid exhibited the highest contact angle with distilled water, indicating the highest degree of hydrophobicity. Among the methods employing butyl acrylate alone, the B-30 condition demonstrated the greatest hydrophobicity. When the surface was modified with maleic anhydride, no significant differences were observed, although the BM-4 sample exhibited slightly lower hydrophobicity. The contact angles measured using diiodomethane were generally similar, except for Nano-Whisker, possibly due to the analytical conditions used during contact-angle measurement.

Surface elemental composition and morphology of the modified nano-whisker were analyzed using SEM-EDS and XPS analyses. SEM images revealed the preservation of nano-whiskers’ acicular morphology under all conditions. The EDS analysis indicated that modifying the filler with lauric acid resulted in the highest reduction in surface Mg content, attributed to superhydrophobicization. The XPS analysis confirmed the decreasing trend of Mg1s content, lauric acid < butyl acrylate < maleic anhydride, aligning with the copolymerization mechanism of small monomers with Mg ions and hydroxyl groups on the nano-whisker surface.

FT-IR analysis was employed to investigate the chemical composition of the modified nano-whisker surface. While the peak changes associated with different surface modification methods and conditions were not remarkably significant, the presence of newly formed chemical composition on the surface compared with the unmodified inorganic filler was confirmed. 

PP composites with surface-modified nano-whiskers were prepared. The migration behavior of inorganic fillers after thermally accelerated aging, according to each modification method, was analyzed using SEM-EDS and FT-IR. The results demonstrated a significant decrease in the extent of inorganic migration upon the inclusion of whiskers, attributed to their enhanced compatibility with the PP matrix. The analysis of the mechanical properties revealed that the surface modification of nano-whiskers effectively improved their compatibility with PP, thereby enhancing mechanical properties under standard conditions and during thermally accelerated aging. 

The results of these investigations imply the potential reduction in aging effects on PP composites, thus offering potential advantages for their application in automotive interiors. While diverse modification conditions have been recognized in this research, the subsequent objective lies in conducting an exhaustive analysis to ascertain the optimal conditions.

## Figures and Tables

**Figure 1 materials-16-05899-f001:**
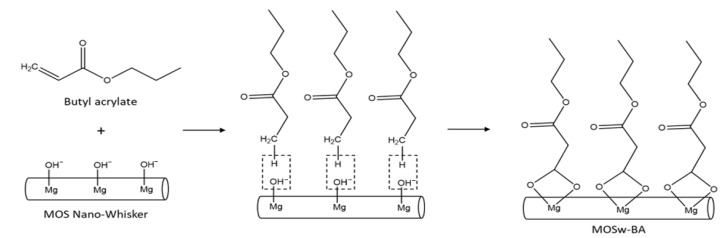
Mechanism of surface modification of MOS nano-whisker using butyl acrylate.

**Figure 2 materials-16-05899-f002:**
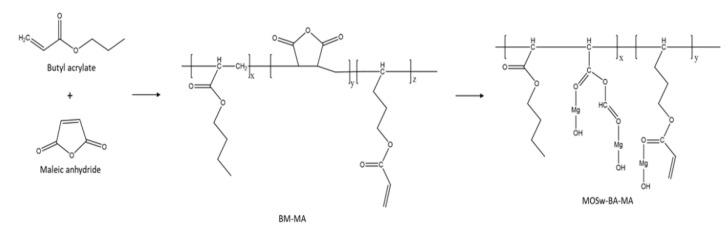
Mechanism of surface modification of MOS nano-whisker using butyl acrylate and maleic anhydride.

**Figure 3 materials-16-05899-f003:**
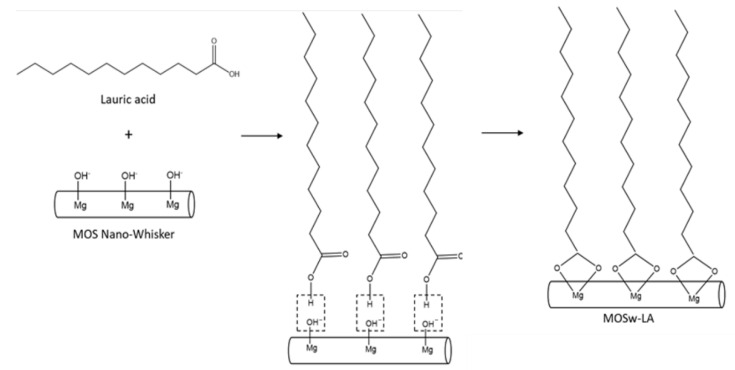
Mechanism of surface modification of MOS nano-whisker using lauric acid.

**Figure 4 materials-16-05899-f004:**
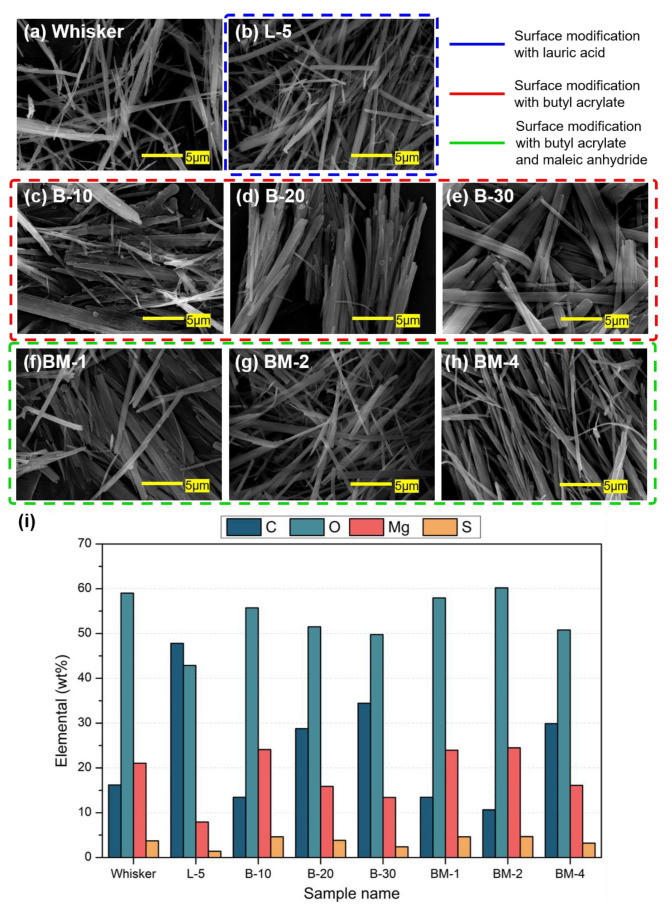
(**a**–**h**) SEM images and (**i**) EDS graph of the inorganic filler’s surface according to the surface modification conditions.

**Figure 5 materials-16-05899-f005:**
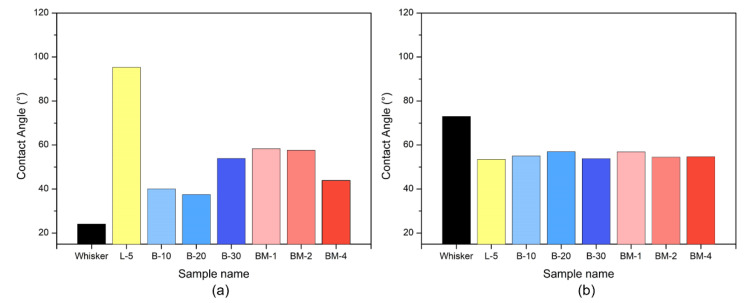
Contact angle of nano-whisker pellets. (**a**) Measured with distilled water. (**b**) Measured with diiodomethane.

**Figure 6 materials-16-05899-f006:**
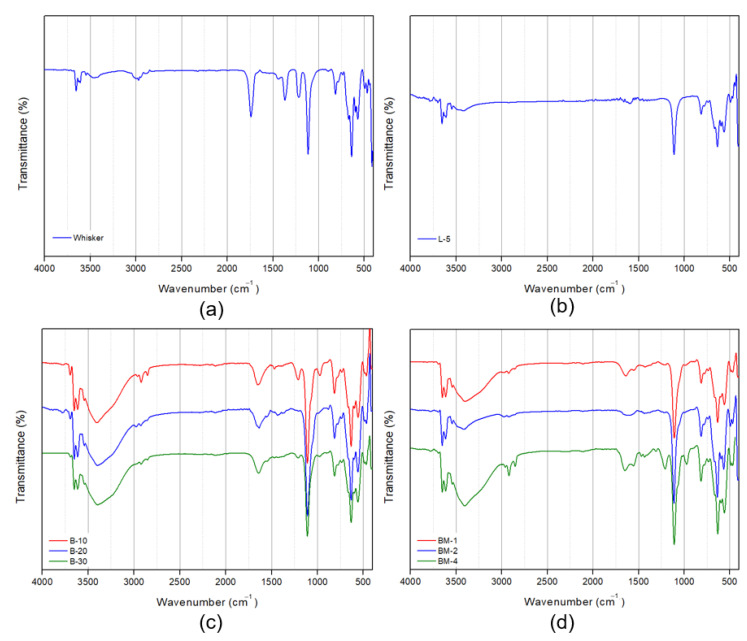
FT-IR graphs of nano-whiskers according to surface modification conditions. (**a**) Whisker; (**b**) L-5; (**c**) B-10, -20, -30; (**d**) BM-1, -2, -4.

**Figure 7 materials-16-05899-f007:**
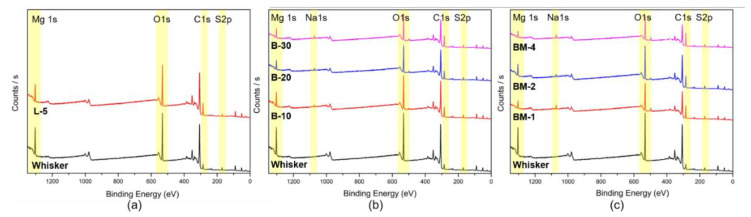
XPS survey graph of nano-whisker. Non-modified nano-whiskers were compared with surface-modified ones obtained with surface modification: (**a**) L-5; (**b**) B-10, B-20, B-30; (**c**) BM-1, BM-2, BM-4.

**Figure 8 materials-16-05899-f008:**
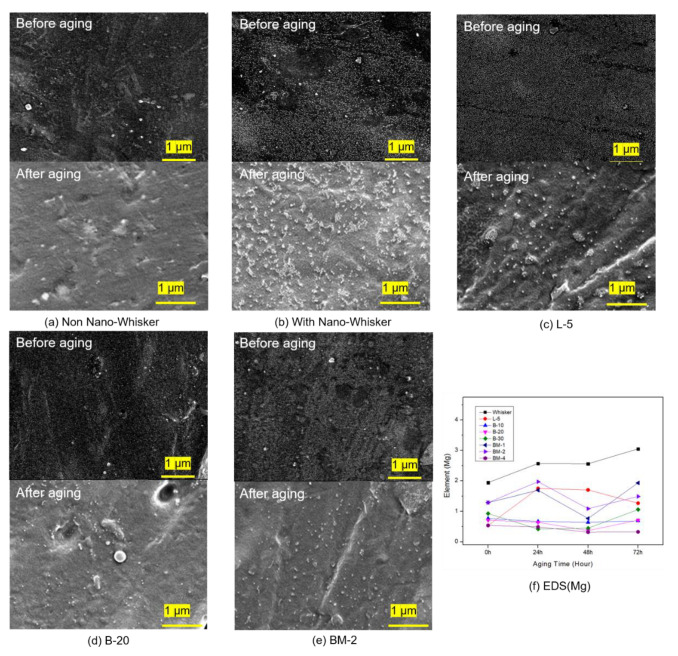
SEM images of the surface of polypropylene specimens after accelerated thermal aging at 140 °C for 72 h. (**a**) Non-Nano-Whisker; (**b**) with nano-whisker; (**c**) L-5; (**d**) B-20; (**e**) BM-2. (**f**) EDS analysis of surface Mg element composition in polypropylene specimens during accelerated aging.

**Figure 9 materials-16-05899-f009:**
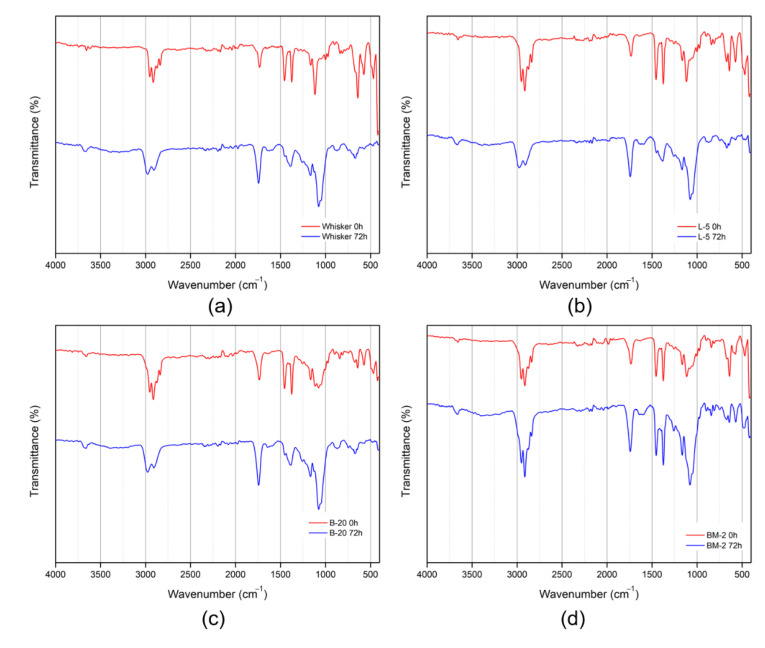
FT-IR graphs of a polypropylene specimen after 72 h of accelerated thermal aging. (**a**) Whisker; (**b**) L-5; (**c**) B-20; (**d**) BM-2.

**Figure 10 materials-16-05899-f010:**
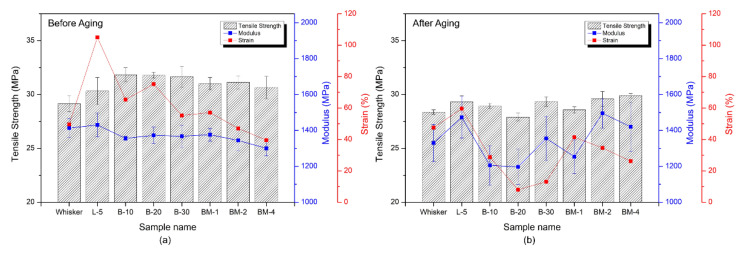
Tensile strength and modulus graphs of tensile test of polypropylene specimens before and after accelerated thermal aging. (**a**) Before thermal aging; (**b**) after thermal aging.

**Table 1 materials-16-05899-t001:** Conditions of nano-whisker surface modification.

Experimental Method	Material	Weight Content	Material	wt%	Sample Name
Method 1	SDS solution	15 wt%	Butyl acrylate	10 wt%	B-10
				20 wt%	B-20
	Nano-whisker	330 g		30 wt%	B-30
Method 2	SDS solution	15 wt%	Maleic anhydride	1 wt%	BM-1
	Nano-whisker	330 g		2 wt%	BM-2
	Butyl acrylate	20 wt%		4 wt%	BM-4
Method 3	Nano-whisker	330 g	Lauric acid	5 wt%	L-5

## Data Availability

No new data were created or analyzed in this study. Data sharing is not applicable to this article.

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
