# Peer review of "Study on the Migration Behaviors of Magnesium Oxysulfate Nano-Whiskers in Polypropylene Composites with Surface Modification"

_materials, 2023, doi:10.3390/ma16175899_

Round 1

Reviewer 1 Report

Won and coworkers reported the hydrophobic surface modification of the hydrophilic inorganic filler through the organic compounds to improve the performance and adhesion of the polymer matrix. The composite characterization and surface test are well conducted to support the hypothesis and conclusion. After addressing the following minor concerns, this work is recommended for publication. 

The discussion of the performance comparison after modifying with lauric acid, butyl acrylate, and maleic anhydride is lacking. The authors should give more explanation on the performance comparison with different composites. 

Reviewer 2 Report

1The article presents the results of MOS modification by using different organic agents, in order to increase compatibility and reduce migration within the polymer matrix. However, significant corrections are required before publishing:

11.      Avoid abbreviations in the title

22.    Define MOS before using abbreviation in Experimental

33.    Did you use previously published procedures for surface modifications? If so, put the references. If not, explain the used concentrations of organic modifiers.

44.    Move the sentence in line 191 before quoting Figure 4b-h

55.    Line 201: confusing part of the sentence ‘and is superhydrophobicization.’

66.    SEM before aging should be included

77.    In line 298, the authors claim that BM4 has the most effective 298 prevention of inorganic filler migration. The SEM of BM4 must be included.

88.    FTIR of composites before and after aging must be included

99.    Explain why did the authors choose NIVEA sunscreen protection, could it be identified with FTIR?

110.   Avoid references in the conclusion, write it in a more concise way

I have entered the correction in the comments to the authors - confusing sentence in line 201.

Reviewer 3 Report

This study focuses on a study of the surface modification of a hydrophilic inorganic filler with hydrophobicity for the reason of to improve the interfacial adhesion with PP. The topic of the paper is actual. The paper is well-structured and written, the achieved results are presented clearly. While I believe the paper to be of high quality, I do have a few comments and suggestions:

1, How were the three methodologies described in section 2.2 of the paper selected?

2, Please provide the appropriate standard for the accelerated aging test conditions. Temperature of 140 degrees Celsius is not typical. In what equipment were the accelerated aging experiments conducted?

3, I recommend using identical graphics for graph bars (Fig. 4 is colored, whereas Fig. 5 is not).

4, Less than fifty percent of cited sources were published within the past five years. I suggest using mainly actual scientific papers.

Based on the comments mentioned above, I suggest a minor revision of the paper.

Reviewer 4 Report

This work presents "Study on the migration behaviors of MOS Nano-Whiskers in polypropylene composites through surface modification". The process achieved in this review is important to this journal. However, as per the discussion part of this manuscript needs more improvement. This manuscript is reconsidered after including and addressing the below listed comments with Major corrections.

Comments for Author:

·       Abstract requires more technical achievements from the proposed works compared to highlight the novelty of the manuscript.

·       The authors need to reduce the abstract content by focusing main findings of the proposed sensor.

·       “Keyword” should not be identical to the manuscript's title and short keywords length.

·       The authors should write the complete terms of all abbreviations (including the instruments) before the first use in the abstract and entire manuscript.

·       The introduction is well structured in a good manner, and most of the things are well supported by the topic and the objectives of the study. The explanation is most fitted with the paper title. However, the author advised describing the innovation and advances concerning the literature and more details on sample preparation.

·       Data, Results and Discussion section; data representations were in informative way. However, under the results and discussion section, I noticed that the discussion section should be rewritten carefully with previous studies and relevant other explements to support your justifications. This is the main flawed thing I noticed and please rearrange the entire results and discussion section based on the way of writing proper discussion.

·       Authors very rarely provide a critical and well-designed experimental comparison. The Introduction is repetitive. In this current version, it seems that everything is good and perfect, which is surely not the case.

·       Authors did not explain about the limitations of the sensor should I assume that there are not limitations? It would be nice if they say the future perspectives and its limitations which can attract more readers.

·       The use of subjects such as "we, they" should be avoided, and the passive voice should be preferred as much as possible.

·       Conclusions. This section should include A summary of your key findings and your vision for future work.

·       This manuscript contains spelling typos, style errors, and grammatical errors, which severely affect readability. Please carefully check the whole manuscript and correct them.

 Minor editing of English language required

Round 2

Reviewer 2 Report

The authors have corrected everything in accordance with reviewer's suggestions.

Reviewer 4 Report

Authors have addressed all the comments in good manner. Now it can accept without further reviews